# Development and Use of the ‘SENS’-Structure to Proactively Identify Care Needs in Early Palliative Care—An Innovative Approach

**DOI:** 10.3390/healthcare7010032

**Published:** 2019-02-20

**Authors:** Monica C. Fliedner, Geoffrey Mitchell, Daniel Bueche, Monika Mettler, Jos M. G. A. Schols, Steffen Eychmueller

**Affiliations:** 1University Centre for Palliative Care, University Hospital Bern, 3010 Bern, Switzerland; monica.fliedner@insel.ch; 2Department of Health Services Research, School CAPHRI (Care and Public Health Research Institute), Maastricht University, Duboisdomein 30, 6229 GT Maastricht, The Netherlands; jos.schols@maastrichtuniversity.nl; 3Primary Care Clinical Unit, Faculty of Medicine, University of Queensland, Brisbane 4072, Australia; g.mitchell@uq.edu.au; 4Centre for Palliative Care, Kantonsspital St.Gallen, CH-9007 St.Gallen, Switzerland; daniel.bueche@kssg.ch (D.B.); monika.mettler@kssg.ch (M.M.)

**Keywords:** chronic life-limiting condition, palliative care, patient-centered care, early palliative care intervention, advance care planning, needs assessment

## Abstract

Anticipatory planning for end of life requires a common language for discussion among patients, families, and professionals. Studies show that early Palliative Care (PC) interventions based on a problem-oriented approach can improve quality of life, support decision-making, and optimize the timing of medical treatment and transition to hospice services. The aim of this quality-improvement project was to develop a pragmatic structure meeting all clinical settings and populations needs. Based on the Medical Research Council (MRC) framework, a literature review identifying approaches commonly used in PC was performed. In addition, more than 500 hospital-based interprofessional consultations were analyzed. Identified themes were structured and compared to published approaches. We evaluated the clinical usefulness of this structure with an online survey among professionals. The emerged ‘SENS’-structure stands for: Symptoms patients suffer from; End-of-life decisions; Network around the patient delivering care; and Support for the carer. Evaluation among professionals has confirmed that the ‘SENS’-structure covers all relevant areas for anticipatory planning in PC. ‘SENS’ is useful in guiding patient-centered PC conversations and pragmatic anticipatory planning, alongside the regular diagnosis-triggered approach in various settings. Following this approach, ‘SENS’ may facilitate systematic integration of PC in clinical practice. Depending on clearly defined outcomes, this needs to be confirmed by future randomized controlled studies.

## 1. Introduction

Anticipatory planning for the end of life, including Advance Care Planning (ACP) as a component of Palliative Care (PC), has become a core element in western healthcare systems to improve the care for people with advanced life-limiting illnesses [1]. Early integration of PC into the illness trajectory means initiating a conversation on future treatment and care wishes, taking into consideration patients’ values and personal circumstances, as well as their social surroundings. In addition, clinicians can plan for anticipated clinical complications that may not be present now but may occur in the future. Finally, the physical, social and emotional needs of the informal carers of patients also need to be considered [2,3]. Such anticipatory planning seems to improve quality of life [4], optimize the timing of treatment and transition to hospice services, and may reduce health care costs [5,6]. Early PC (ePC) is best incorporated into care pathways of chronic life-limiting diseases in need of a PC approach alongside the traditional diagnosis-focused approach by initiating a conversation on future treatment and care wishes [7]. This approach is suitable for patients with any chronic life-limiting disease, such as patients with advanced cancer [8,9] or advanced chronic kidney disease contemplating haemodialysis [10], as well as patients already on haemodialysis with severe co-morbidities or those considering withdrawing from dialysis [11] or suffering from Amyotrophic Lateral Sclerosis [12] or advanced heart disease [13]. A major challenge is finding a common language for discussions on future care and ACP among patients, families, and health care professionals [14].

However, the most effective but still easy to use thematic structure in an ePC intervention to identify and prioritize areas used in an ePC intervention [15] and how to prioritize care needs without overburdening patients and carers is still unknown. For clinical purposes, our team of Swiss PC professionals identified the need for an easy-to-use structure, which could support active participation of patients and their families in conversations with professionals to prioritize care and anticipate care planning. The aforementioned professionals, working in a hospital-based specialized PC-consultation service in St.Gallen and Bern, consisted of two physicians and two nurses, all trained in PC at the specialist level. In addition, a sociologist worked during the time of the development of the structure on her Ph.D. in PC, including communication and interaction with people at the end of life. The structure should help patients to become more self-efficient by subdividing the current or future challenges into “manageable proportions” [16]. It might also assist patients to make carefully considered decisions in various domains, including future medical care.

We present how our team developed the so-called ‘SENS’-structure (symptoms, end-of-life decisions, network, and support)—a thematic structure for anticipatory planning as the essence of ePC interventions. The purpose of our PC hospital-wide, quality-enhancing project was to develop a pragmatic and, by the interprofessional team, concise assessment structure to identify patients’ major concerns and priorities, as well as their main resources. The structure should (a) be easy to use, (b) support mutual agreed multidimensional care plan to facilitate self-efficacy in medically challenging situations, (c) facilitate the evaluation of interventions toward defined goals of care, and (d) ease the information flow between general and specialized PC. Furthermore, it should eventually serve as an educational structure for patients and health professionals, as well as support financial reimbursement for PC interventions.

We explore the clinical value and implementation of the ‘SENS’-structure and its usefulness for planning, documentation, evaluation, and education primarily in the acute clinical setting from the perspective of professionals. We are going to discuss its uptake, challenges, and opportunities. Its validity and effects in the interaction with patients will be the subject of future articles.

## 2. Materials and Methods

For the development of the ‘SENS’-structure (Figure 1), we used the Medical Research Council’s (MRC) 2000 framework for developing complex interventions [17], which we followed in five distinct phases: a preclinical phase, a modelling and piloting phase, which was followed by an explorative phase. To show all developmental phases of the intervention, the subsequent research phases will only briefly be mentioned.

As part of the hospital quality improvement project in St.Gallen and Bern [18], we followed a cyclic and reflective action–research approach, primarily including feedback from professional users for the pre-clinical phase, as well as phases 1 and 2. For the retrospective chart review in phase 1, ethical approval was not required. For the survey among professionals (phase 2), ethical approval was obtained from the local ethical committee.

Pre-clinical phase (2008–2010): Based on an exploration of the literature, we tried to identify clinical approaches that were commonly used to structure care-planning conversations thematically. We used the search terms “palliative care”, “needs assessment”, “assessment of health care needs”, “guideline health planning”, and “advance care planning” using the Boolean operators “and/or”, respectively. Our interprofessional team of PC experts critically evaluated these approaches for their clinical application and practicability. Derived common goals of PC as defined within the Swiss national PC strategy [19] served as a first structure.

Phase 1: During the modelling phase (2010–2012), we tested the feasibility and acceptability of the first structure in daily clinical practice. For that, we systematically reviewed, retrospectively, more than 500 records of documented inpatient PC consultations, performed by our interprofessional PC team with patients disregarding their disease and their families. We compared and ordered clusters of common themes into the goals of PC [20] and discussed any discrepancies among our team until we reached consensus. Through this process, we developed the ‘SENS’-structure.

Phase 2: During the exploratory phase (2012–2018), we clinically tested the usefulness of the ‘SENS’-structure for initial PC assessment, organization of the interprofessional care plan, documentation, and evaluation of patient preferences and priorities. Over a two-year period, and as part of the PC national audit process [21], professionals from other settings who also used the ‘SENS’-structure in their clinical practice after its publication [20] provided us with feedback, which we considered in the final refinement of the ‘SENS’-structure. In this period, we observed how ‘SENS’ was used and implemented in different clinical settings, as well as by educational and political organizations.

In addition, in October/November 2018, we performed an online survey among clinical users of the ‘SENS’-structure in the German speaking part of Switzerland in various settings (hospital, home care, long-term care) to collect formal and anonymized feedback from professionals. The local ethical committee of Bern considered this survey as being outside their sphere of responsibility. Participants (collaborating professionals, such as general practitioners, PC physicians, and nurses from hospitals and former students of the specialized PC training) were identified and approached by our two major PC centers (University Hospital in Bern and at the Cantonal Hospital St.Gallen), providing a link by email to the respective survey. Their anonymous participation was considered as consent to participate in the survey.

In phase 3, we performed a randomized controlled ‘SENS’-study [22] (2014–2017) in patients with advanced cancer, including an integrated qualitative study exploring patients’ experience with a ‘SENS’-based intervention and reflecting upon the ‘SENS’-themes for their consistency and completeness.

In phase 4, we are going to study the long-term implementation of the ‘SENS’-structure in outpatient care through a cluster randomized controlled trials (RCT). Within this article, we focused only on the developmental and explorative phases and will discuss the findings of Phase 3 and 4 in later publications.

## 3. Results

### 3.1. Pre-Clinical Phase

As major thematic structures for anticipatory planning in PC, we identified the definition of the World Health Organization (WHO) on PC [23,24], the listing of PC needs from the National Comprehensive Cancer Network (NCCN) Guidelines on PC [25,26], and the “PEPSI-COLA” structure within the National Gold Standards Framework (GSF) [27,28,29]. All structures had certain goals of care in common which were identified and clustered alongside the four goals of the Swiss national PC strategy: (a) Improving self-efficacy and self-help capacity, (b) promoting self-determination by supporting a certain sense of coherence in decision-making, (c) ensuring safety in sometimes life-threatening situations, and (d) assuring support of the encumbered carer including the bereavement phase. Subsequently, the topics covered by these guidelines were assigned to the identified goals of PC (Table 1).

### 3.2. Phase 1: Piloting and Modeling Phase

By analysis of the content of consultations, we identified four main clusters of themes that we regrouped to form the ‘SENS’-structure. The acronym ‘SENS’ represents the following themes (Table 1 and Table 2):Symptoms patients suffer from or worries they may have, including self-support strategies in case of a crisis, as well as carer empowerment in symptom crises;End-of-life decisions in regard to the future, including individual care and treatment preferences, potentially formulated as an advance directive;Network organization (private and professional), including assessment of current living circumstances and organization of support in case of an emergency not manageable at home/nursing home;Support for the carers to cope with the situation and to prevent overburdening.

We compared the goals of care identified in the preclinical phase with the ‘SENS’-structure and saw that ‘SENS’ seems to covers all relevant themes. Subsequently, we used ‘SENS’ to design a patient prompt sheet with concrete questions to assist patients and families in preparing for ePC conversations with health professionals (Table 2).

### 3.3. Phase 2: Exploratory Phase

Together with other Swiss healthcare facilities, we evaluated the usefulness of ‘SENS’ in the acute hospital and outpatient PC setting.

A formal evaluation among professionals revealed important feedback. The majority (49.1%) of the respondents had a nursing background (Table 3), working either in a specialized hospital-based or home care service, with a considerable amount of PC experience.

Overall, ‘SENS’ was used by the respondents both in the early PC phase and in the dying phase, respectively (72.2%). Respondents used ‘SENS’ mainly within the interprofessional team context, and it was rated most helpful for providing an overview of current problems and individual expectations, as well as for interprofessional collaboration and coordination. The vast majority of the respondents (81.5%) felt that the active participation of patients and family carers in defining a common care plan was well supported by the ‘SENS’-structure. Although professionals need certain training to lead the conversation with the patient according to the ‘SENS’-structure, most participants (87%) had good experience with the use of ‘SENS’. Specifically, when patients do not want to address the issue of dying and death, some professionals felt that it was challenging to use ‘SENS’ with all its components.

Feedback from professionals (nurses and physicians) and patients as part of annual medical quality assurance audits, as well as responses from the survey, revealed benefits in the following four main areas:

**Firstly,** the ‘SENS’-structured initial assessment seemed to help patients and families to gain a systematic overview of future challenges. ‘SENS’ encouraged them to talk about these—often very personal—worries. Often, a medical diagnosis-driven approach was in the foreground; therefore, some patients seemed challenged by focusing early on non-medical topics and analyzing their situation systematically.

We found that all four areas of ‘SENS’ were equally relevant. ‘SENS’ provided an initial overview which could then be assessed in greater detail using validated assessment tools for specific symptoms whenever appropriate. The structure seemed to be evenly helpful in ePC, in more advanced and complex palliative situations and the dying phase. The distinction between managing symptoms and discussing end-of-life decisions was obvious to clinicians. In some documented conversations, the difference between the “*N*” = network (who belongs to the social network and which professional systems do the patients and families have in place) and the second “*S*” = support for the carers was not always clear. We concluded that “support” as term alone may be misinterpreted and added “for the carers” to specifically focus on the assessment of carer or family burden.

These findings were supported by the results of the online survey, in which the majority of the participants felt that all four areas covered always or almost always the situation of patients. This was particularly clear in the themes ‘Symptom management’ (94.6%), ‘End-of-life decisions’ (89.3%), and ‘Network-organization’ (78.6%). Probably the theme ‘Support for the carer’ still needs more refinement since 50.9% of the survey participants felt that it covered the support for patients but not the support needs for the burdened carers. It offered a needs-based approach for multifaceted situations driven by concrete problems of daily life, while utilizing resources of patients.

From the perspective of health care professionals, ‘SENS’ has been independently used and proven to be helpful in clinical practice [30] for structuring the assessment of the main concerns. It appeared to be helpful as a “short and simple” enough assessment structure for general and specialist PC settings [19,31], such as hospitals (inpatient PC, consultancy services), nursing homes, or primary care settings.

**Secondly,** from clinical practice, we know that ‘SENS’ was often used in interprofessional rounds. It showed that ‘SENS’ served as a useful structure to develop a care plan together with patients, guiding the setting of goals of care, expectations, prioritization, responsibility, planning, and allocation of resources and financial reimbursement. It also supported the collaboration within the interprofessional team (Table 4).

Clear distribution of tasks and responsibilities within the interprofessional team, including timelines and changes of priorities, could be outlined from the beginning of the PC intervention based on ‘SENS’. As an example: if physical symptoms were initially in the foreground, close cooperation between the medical and nursing staff was necessary. The focus could later shift to discussing the future place of care, including network organization and emergency planning, for which the social worker and mobile PC teams may have taken the lead.

**Thirdly,** ‘SENS’ structured the evaluation of care, which allowed a better judgment of the complexity within a particular area. Based on a common language, ‘SENS’ offered an effective way to prepare and document a problem-oriented family conference in which the interprofessional team members each had their specific tasks. In addition, ‘SENS’ helped to thematically structure case discussions and critical review.

**Fourthly,** ‘SENS’ has lately been officially recommended as a structure for professional education and reimbursement. The areas of ‘SENS’ can be weighted depending on the target of professional education (physicians, nurses, social workers, etc.), as shown in postgraduate basic and specialist PC training programs in Switzerland [32]. At the same time, it can ensure that all essential topics are equally covered. Since 2015, the ‘SENS’ structure was officially implemented in Swiss medical schools by the publication of Eychmueller et al. [33] as recommended teaching material. In addition, ‘SENS’ was adopted as a basic assessment-structure triggering the comprehensive diagnosis related groups (DRG)-codes for complex PC treatment within the SWISS Health Care System [34].

### 3.4. Phases 3 and 4: Patient-Centered Research and Implementation

In phases 3 and 4, we are currently investigating the impact of a ‘SENS’-structured conversation on patients’ distress and trust through two RCT. In a prospective RCT including an embedded qualitative study with ePC cancer patients (ClinicalTrials.gov Identifier NCT01983956) [22], we investigated whether ‘SENS’ is helpful to increase patient self-efficacy and reduce distress and costs, as suggested in other studies. These results are in submission or under review for publication, respectively [35,36]. To create additional evidence for the usefulness of ‘SENS’ in the primary care setting, a cluster RCT [37] is underway.

## 4. Discussion

The ‘SENS’-structure was developed using a stepwise approach for complex interventions, as recommended by the MRC [17]. Most of the existing guidelines did not seem designed for clinical use [24] or were highly complex [26] or needed further refinement for the use in hospitals and across care settings [38]. In contrast, ‘SENS’ seemed to meet the expectations of professionals for assessment, planning, evaluation, education, documentation, and communication in various settings across the care continuum. Other structures [39] or frameworks were either not designed for assessing care needs or considered rather complex in non-specialist PC settings and were, therefore, not taken into consideration. As part of the Swiss national PC strategy, the ‘SENS’-structure was rapidly adopted for clinical and educational purposes because of its simplicity and comprehensiveness.

### 4.1. Impact of the Use of the ‘SENS’-Structure on Clinical Practice

In terms of completeness compared to other assessment structures in PC, from the health care professionals’ perspective, ‘SENS’ seemed to covers all relevant themes. Clinical audits showed that ‘SENS’ seemed to support patients and carers to be an active part in generating anticipatory planning and to feel empowered for future challenges. In addition, it seemed to enable interprofessional teams to prioritize and focus, together with patients and families, on what is important and manageable in individual situations.

Despite complex medical situations, ‘SENS’ seemed to help patients to regain control over personal areas of life and to shift the focus to patients’ strengths and resilience, facilitating their self-efficacy. More research on this effect has been undertaken. Thus, ‘SENS’ is more likely to contribute to a flexible and equally comprehensive concept of biopsychosocial and spiritual care, alongside diagnosis-based medical reasoning and disease-specific approaches, as recently proposed through the term “concurrent care” by the American Society for Clinical Oncology (ASCO) [40,41]. Concurrent care with ‘SENS’ as the patient-centered assessment structure may foster the integration of PC in other vulnerable populations, such as people with dementia [42,43] or other chronic life-limiting diseases, which is in line with the World Health Assembly’s call for action [44]. 

### 4.2. Effects of the Initiation of a ‘SENS’-Based Conversation

During the different phases of developing and piloting ‘SENS’, we encountered clinical acceptance by professionals when ‘SENS’ was administered alongside medical reasoning.

No harmful effects of the ‘SENS’-structure have been mentioned in any feedback from professionals, but more detailed evaluation is underway. Some professionals would expect that patients could experience the systematic analysis of ‘SENS’ themes as challenging, perhaps because patients might not be prepared to deal with the life-limiting nature of their disease or to discuss PC issues early on in their disease trajectory. This could especially be true in a health system with a clear focus on “cure”, with healthcare providers being less likely to actively and early address PC needs. ‘SENS’ may help to structure communication on anticipatory planning complementary to medical reasoning. Health professionals may finally feel better prepared and trained for end-of-life conversations. Such conversation has earlier been entitled ‘the multimillion dollar conversation’ because of its ability to save money through the redirection of hope and trust from potential toxic medical intervention to individual goals of care [6].

### 4.3. Impact of ‘SENS’ on Financial Reimbursement of PC Interventions

The current Swiss healthcare system is a strongly diagnosis-oriented system (DRG), while PC is a problem-based approach. Defining specific activities for PC, therefore, is paramount to obtain sufficient financial reimbursement. The ‘SENS’-structure has recently been officially acknowledged as the basic assessment-structure triggering the comprehensive DRG-codes for complex palliative treatment [34]. Introducing a problem-based structure like other ePC interventions [45,46], the ‘SENS’-structure could have a positive impact on overall health care costs in the last months of life, with more emphasis on problem-solving than expensive diagnosis-driven interventions.

### 4.4. Impact of ‘SENS’ on PC Education and Research

The ‘SENS’-structure has been integrated into professional PC curricula at undergraduate and postgraduate levels in our country [33]. The four themes of ‘SENS’ seemed to cover what PC is actually offering and doing in clinical practice, providing an overview which is easy to understand—for professionals, as well as for the broad public. Of course, conversations based on ‘SENS’ need to be trained, especially in the ways how to speak about certain sensitive topics at the right time to not overburden the patient.

Although qualitative and health service research is getting more attention, areas such as decision-making, health care services/network, and areas of burden and resources of the family are less frequently a subject of interprofessional research. ‘SENS’ may also allow for a more comprehensive view on palliative research topics and on their weighting.

Further development and refinement of the ‘SENS’-structure and testing in clinical and educational settings through research is needed. Whether ‘SENS’ truly covers all aspects of assessment and care planning in PC for specific populations, such as pediatric, elderly, and patients with dementia, remains to be determined by systematic research.

## 5. Conclusions

Since the medical field is mostly aligned with a diagnosis-focused approach, a problem and resources-oriented approach is rather unusual. ‘Concurrent care’ has recently become a promising model of care to bring these two approaches together. The ‘SENS’-structure was developed to foster a strong patient-needs-based and participatory approach in such a model.

The starting point of our search for a meaningful person-centered interprofessional PC approach led us toward a structure that is easy to use in clinical practice in and across all settings, but also in PC research and professional education. Support from a specialized PC team might encourage and facilitate the implementation of the ‘SENS’-structure into care pathways for patients with a life-limiting disease.

The ‘SENS’-structure seemed to covers all themes and complexity as proposed by other structures, such as the GSF or the NCCN guidelines for PC. Providing guidance for ePC conversations, including ACP, it focuses particularly on patient perspectives. Practice experience so far shows that ‘SENS’ helped patients and their families facing complex PC situations to plan proactively and to be prepared for future challenges, including death and dying. In addition, professionals reported its usefulness to prioritize and focus on what is important and manageable in the individual situation and to define task distributions within a common care plan through direct cooperation and mutual respect. Further research to confirm the usefulness of the ‘SENS’-structure as experienced by patients and clinicians is required in phases 3 and 4 of the MRC framework.

In the German-speaking part of Switzerland, the ‘SENS’-structure has been implemented in clinical practice in hospitals, as well as in primary care over the past six years and has been recommended by the Swiss federal office of health. We anticipate that the ‘SENS’-structure may play a significant role in further development of ‘concurrent’ treatment, combining disease-modifying treatment and a person-centered care plan, making sense for patients’ family, and carers, as well as for professionals.

## Figures and Tables

**Figure 1 healthcare-07-00032-f001:**
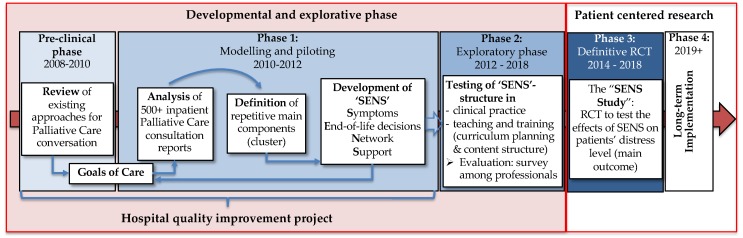
Phases of development the ‘SENS’-structure based on the Medical Research Council (MRC) framework (2000).

**Table 1 healthcare-07-00032-t001:** Overview of themes discussed in Palliative Care (PC) conversations.

Common Goals of PC (Switzerland, 2014)	WHO ^1^ Definition of PC (2002)	NCCN ^2^ Guideline for PC (2009/2016)	GSF ^3^ (2014)	Theme	‘SENS’-Structure Theme (Eychmueller, 2012)
Improving Self-help capacities				Self-effectiveness	Symptoms
	×	×	×	Physical	
		×	×	Educational and informational needs	
	×		×	Spiritual	
	×		×	Quality of life	
			×	Personal (e.g., inner journey)	
			×	Late (e.g., death rattle, agitation)	
Promoting Self-determination		×	×	Self determination	End-of-life decisions
		×		Benefits and risks of (anticancer) treatment	
			×	Emotional (e.g., fears, relationships)	
	×			Dying issues	
Ensuring Safety in sometimes life-threatening situations			×	Out of hours—emergency	Network
				Safety	
	×	×		Psychosocial	
		×		Cultural factors affecting life	
Assuring Support for the encumbered family	×		×	Support of family	Support of the carer
			×	Afterwards (bereavement period)	

Abbreviations: ^1^ WHO: World Health Organization; ^2^ NCCN: National Comprehensive Cancer Network; ^3^ GSF: Gold Standards Framework^®^.

**Table 2 healthcare-07-00032-t002:** Patient prompt sheet: ‘SENS’-structure with themes and potential assessment questions.

‘SENS’—Theme and Definition	Potential Assessment Questions: To Identify Patients’ and Their Family’s Priorities It is Essential to Ask Questions that will Help the Patient to Focus
**Symptom management:** The best possible way to treat the symptoms and to self-empower the patient for self-help in dealing with the symptom.	Which problems, themes, or symptoms are you the most worried about at the moment, and which concerns are you the most worried about for the future? Concerning the topics below, which make you feel anxious? ○ Physical, e.g., pain, nausea, dyspnea, fatigue; ○ Psychological, e.g., limitations in thinking, grief, anger, anxiety, depression; ○ spiritual, e.g., faith/religion, question of meaning, hope and despair; ○ socio-cultural, e.g., own role (family/profession), tradition/rituals, relationships. In which areas did you have good experiences so far? What helped or supported you in these situations? What resources helped you to deal with symptoms, problems, challenges?
**End-of-life decisions and expectations:** Step-by-step and self-regulated decision-making, definition of personal preferences, and preventive planning for potential complications.	How have you made important decisions so far in your life (e.g., alone, support people)? Or, did you mainly rely on the advice of others? Or, did you let others decide for you? What is very important to you? What do you want to urgently experience or resolve? Which goals would you like to achieve (with medical measures)? Which questions regarding your disease are still unanswered? What (and how much) do you want to know? Regarding your dying—what needs to be clarified and managed related to your plans for death? What would you like to determine in advance (e.g., in an advance directive)? What is your attitude toward life-prolonging measures, i.e., Cardiopulmonary Resuscitation-measures in case of a circulatory arrest? What should be done with your body after you die (autopsy, organ donation)? Do you have any special wishes or ideas that should be done for you when you cannot decide anymore for yourself (including care, rituals, funeral)? Is there any unfinished business that you want to deal with or arrange?
**Network-organization:** Professional (including out-of-hours support) and private care network.	If your health situation does not improve substantially—where would you like to stay? What is your home environment like (e.g., stairs, access to the bathroom)? Who of your family members/friends can support you when you become weaker and you lose your strength? Who of your family or friends could you involve in your care? Which professionals (e.g., social services, spiritual care, community nursing, general practitioner, volunteers) are available? If any complication occurs or if there is an emergency: What can you do? What should your family do? Who should be involved in your chain-of-rescue? What are your alternatives for any future care (e.g., nursing homes)? Do you need to consider and maybe already plan ahead in case the care at the location of your choice (e.g., being as long as possible at home) is not possible?
**Support of the carers:** support system for family members, including in the bereavement phase, and for the involved professional carers.	Who of your main family members or friends will most likely need support? Who is already available for support? Which professionals, family members, or friends can offer support? Who will provide support after your death to your family and friends? Does your family need additional specific support (e.g., in financial or legal matters)?

**Table 3 healthcare-07-00032-t003:** Demographics.

Variables		Number	Percentage
Gender (*N* = 55)	Male	18	31.6%
	Female	37	64.9%
	Unknown	2	3.5%
Years in profession (*N* = 56)	Professional years in general (mean)	19.4	
	Years working in PC (mean)	7.6	
Profession (*N* = 57)	Physician	26	45.6%
	Nurse	28	49.1%
	Pastoral carer	2	3.5%
	Psycho(onco)logist	1	1.8%
Main place of work (*N* = 57)	Hospital-based general PC service	7	13.0%
	Hospital-based special PC service	21	38.9%
	General practitioner	9	16.7%
	Home care	12	22.2%
	Nursing home	6	11.1%
	Hospice	2	3.7%

**Table 4 healthcare-07-00032-t004:** Interprofessional team working together with the patient and family based on ‘SENS’.

‘SENS’-Themes	Involved Professionals (Examples)
Symptoms	Physician, nurse, physiotherapist, psychologist, pastoral care worker, dietician, music or art therapist.
End-of-life decisions	Physician, nurse, pastoral care worker, psychologist.
Network-organization	Nurse, social worker, general practitioner, volunteer service.
Support of the carer	Physician, nurse, psychologist, spiritual counselor, social worker.

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
