# Peer review of "Development and Use of the ‘SENS’-Structure to Proactively Identify Care Needs in Early Palliative Care—An Innovative Approach"

_healthcare, 2019, doi:10.3390/healthcare7010032_

Round 1

Reviewer 1 Report

The authors describe development of SENS, a clinical tool for facilitating timely introduction of difficult conversations and involvement of palliative care services at the end of life. The tool is the subject of a four phase study, of which the last two phases remain to be concluded. On the basis of phase 2, which is exploratory, the authors suggest that SENS is acceptable to clinicians and helpful to families and patients.

The authors do not present much data to support those conclusions.  There are no quantitative measures or qualitative outcomes such as quotations from semi-structured interviews.  I found it hard to know the basis on which the authors conclude that the tool has been effective.

One reason is that they have attempted to describe all four phases of this study.

That has left little room for setting out the data.   I would suggest they rewrite the manuscript as a draft report of the exploratory phase (phase 2) as a standalone study, which should include the quantitative and/or qualitative data they accumulated during that phase in support of their conclusion that SENS is a useful tool.

I think the authors are probably right that this is useful tool that will facilitate transition into palliative care, particularly perhaps in the authors’ home country or Switzerland. But rather than describing a complex study which is incomplete and whose findings are not yet clear, it seems to me that it would be better at this point to describe clearly a small study that is complete and has generated data that can support a conclusion.

Author Response

Dear reviewer,

We highly appreciate your comments to help us improve the article. We wrote detailed feedback in the separate pdf file which will hopefully contribute to a better understanding of the article and the development of the 'SENS'-structure for Palliative Care interventions.

Best wishes on behalf of the authors,

Monica Fliedner

The authors describe development of SENS, a clinical tool for facilitating timely introduction of difficult conversations and involvement of palliative care services at the end of life. The tool is the subject of a four phase study, of which the last two phases remain to be concluded. On the basis of phase 2, which is exploratory, the authors suggest that SENS is acceptable to clinicians and helpful to families and patients.

The authors do not present much data to support those conclusions.  There are no quantitative measures or qualitative outcomes such as quotations from semi-structured interviews.  I found it hard to know the basis on which the authors conclude that the tool has been effective.

Response: Thank you very much for your comments – we included some more data that we gained from the survey that we conducted among professionals in Phase 2 to strengthen the conclusion that professionals appreciate the ‘SENS’-structure. The quantitative measures and qualitative feedback from patients are matter of phases 3 and 4 and will be published in future articles.

One reason is that they have attempted to describe all four phases of this study.

That has left little room for setting out the data.   I would suggest they rewrite the manuscript as a draft report of the exploratory phase (phase 2) as a standalone study, which should include the quantitative and/or qualitative data they accumulated during that phase in support of their conclusion that SENS is a useful tool.

I think the authors are probably right that this is useful tool that will facilitate transition into palliative care, particularly perhaps in the authors’ home country or Switzerland. But rather than describing a complex study which is incomplete and whose findings are not yet clear, it seems to me that it would be better at this point to describe clearly a small study that is complete and has generated data that can support a conclusion.

Response: Thank you for this valuable comment - we report on how ‘SENS’ was developed and how it seems to support clinical practice of palliative care. As suggested by the reviewers, the authors focused on the pre-clinical as well as modelling, piloting and exploratory phases of the development of the ‘SENS’-structure (pre-clinical phase and phases 1 and 2). Data of the randomized controlled trial including results from patient interviews will follow in phases 3 and 4 and published later.

New references: 4, 5, 6, 18, 19, 21, 24, 29-34, 38, 42, 43, 45 and 46
Deleted reference: 29

Reviewer 2 Report

Thank you for the opportunity to read the current manuscript, Development and use of the ´SENS´-structure to proactively identify care needs in early palliative care – an innovative approach, which focuses on an important area to enable the integration of palliative care at an early stage in a structured manner

After reading the manuscript, I understand that the structured way has both been tested, evaluated in two RCT studies and implemented in clinical work. It is a solid work that has been done and I look forward to read the results from the RCT studies. To some extent, it comes back to the current manuscript in the form of much being implemented and here is presented the original development and test of the SENS-structure. Something that I ask the authors to take in further consideration as it was somewhat difficult for me in the review phase to handle.

TITLE: The title seems relevant

ABSTRACT: The abstract is informative and present the process clearly. This is to some extent clearer than in the manuscript. Thinking most of the method part. Will come back to this

1: INTRODUCTION: This section is seen relevant regarding its content and with reference to applicable literature.

a.     text on lines 56 and 65 is very similarly formulated and is almost perceived as repetition.

2: MATERIALS AND METHODS

The method part follows a structured form and where figure 1 in a good way helps the reader to follow the different steps for the overarching project. Hence it is a bit difficult to clearly understand what the current manuscript is focusing. Further, to be able to understand how the development and test of the SENS-Structure has been done according to methodological as well as scientifically-ethical aspects. In the latter, how participants were independent from the researchers and had a voluntary participation. Below are presented some parts that may need to be clarified to understand this as a reader.

a.     Pre-clinical phase: here´s a literature search presented as the basis for identifying clinical approaches of thematic structures for care-planning conversations. How this literature review was carried out is not clearly described and in the results section, e.g. the WHO's definition of palliative care and NCCN's guidelines is presented. The question arises whether the authors identified these based on the literature review or basically used well-known documents and guidelines. The discussion then argues that most existing guidelines and documents are not designed for clinics. In fact, I can agree, but wonder if it really is its goal. Here, structured aids in the form of the SENS-structure can be good aids. My wonder is how did the authors reach the guidelines and documents you use? Why, for example, is not The European School of Oncology´s definition of palliative care, or the EAPCs, not included? If those presented in Table 1 were identified in a certain way this needs to be described.

b.     In this pre-clinical phase, a local interprofessional PC expert team evaluated the thematically structures for its usefulness and practicability. To be able to evaluate the process for me as a reader this teams expertise needs to be described. Is this the same team described in line 58? And presented as “we” in Phase 1? And further professionals in line 99? It needs to be clarified who the expert team are and further who the professionals are.

c.     Phase 2: the process within this phase needs to be clarified. As it is described now it poses questions of how this part was performed. Who were involved, including both collaborating PC consultant services? Which were the different settings? How many times were the SENS-structure used? And how was the feedback given? These aspects remain unclear also reading the results from this phase.

d.     Phase 3 and 4 are presented briefly and that they should be presented in other publications. Despite this, parts from these phases are included in the results section and with an online survey that is not presented in the method section, which makes it difficult to really understand what the current manuscript focuses on.

3: RESULTS

The result is presented in continuous text and tables in a structured way. However, the same things that are asked for above remains unclear also here.

1.     The entire manuscript wins on the fact that the aspects already mentioned are considered and clarified and then the result might become clearer. It is of importance that it is made clear, who has been involved and which health care facilities, in order to be able to weigh the discussion against results and the development of the SENS-structure. Phase 3 focused on pts with advanced cancer and this is to my view the only time pts diagnosis is described except from descriptions in the introduction. I know that the specific diagnosis might not be the focus for Palliative care, but as the authors focusing om pts with advanced cancer in phase 3, not mention others or how this is handled, and the conclusions made in line 222-223 I think this need some explanation.

2.     In Table 1 the authors use abbreviations (Sy, E, N, and SU) and numbers in relations to these abbreviations in some places but not in others. It is not clear how to understand these according to the explanation below the table and this needs to be clarified.

3.     In the part starting at line 147, the authors describe a review made of documentation of clinical conversations. I can’t find any explanation how this was performed. Sometimes the documentation might miss different aspects so it would be helpful to have some descriptions about how this was done.

4.     Please check line 161 to165 if correctly written.

5.     The results from phase 4, starting at line 172 describes the SENS- structure to be helpful for inpatient PC and consultant services, and simple to use in various health care settings. It is a bit difficult to value this as no other patient groups but pt with advanced cancer are described and which different health care settings? This needs to be clearified.

4: DISCUSSION

The discussion points out development and use of the SENS-structure covering several years of work, including how the SENS-structure has been developed, implemented in further research and PC curricula at both undergraduate and postgraduate levels, as well as officially acknowledge for use in hospital settings in Switzerland as the basic assessment-structure for ePC. In the earlier parts I have tried to point out some uncertainties that need to be clarified so that readers can form an idea of the secured process in the development and use of the SENS structure. It may well be that the authors who have the results of the two RCT studies clearly understand this, but I do not find it quite clear in this manuscript. Perhaps a larger part of the manuscript should be concentrated on the pre-clinic phase and to include a clear description about in which facilities and for whom the SENS-structure has been tested. Further, I cannot find anything in the result part about test of the SENS-structure in any educational situation although this has already been implemented in PC curricula as well as acknowledge in the Swiss healthcare system as a basic assessment-structure triggering the comprehensive DRG-codes for complex palliative care treatment.

I hope my comments is clear and to some help to advance the manuscript further.

Author Response

Dear reviewer,

We highly appreciate your extensive comments to help us improve the article. We wrote detailed feedback in the separate pdf file which will hopefully contribute to a better understanding of the article and the development of the 'SENS'-structure for Palliative Care interventions.

Best wishes on behalf of the authors,

Monica Fliedner

Thank you for the opportunity to read the current manuscript, Development and use of the ´SENS´-structure to proactively identify care needs in early palliative care – an innovative approach, which focuses on an important area to enable the integration of palliative care at an early stage in a structured manner.

Response: Dear reviewer – we appreciate your comments, we indeed intent to be able to integrate palliative care through this clinically relevant structure SENS.

After reading the manuscript, I understand that the structured way has both been tested, evaluated in two RCT studies and implemented in clinical work. It is a solid work that has been done and I look forward to read the results from the RCT studies. To some extent, it comes back to the current manuscript in the form of much being implemented and here is presented the original development and test of the SENS-structure. Something that I ask the authors to take in further consideration as it was somewhat difficult for me in the review phase to handle.

Response: Yes, this is what the article is about – the development and testing of the SENS structure without going too deep into the final phase of the MRC framework which would be the definitive RCT and long-term implementation – these are on the way but for the completeness of the MRC phases we just want to mention where we are going.

TITLE: The title seems relevant.

Response: Thank you – we will leave it that way.

ABSTRACT: The abstract is informative and present the process clearly. This is to some extent clearer than in the manuscript. Thinking most of the method part. Will come back to this

Response: Thank you – we adapted the method section.

1: INTRODUCTION: This section is seen relevant regarding its content and with reference to applicable literature.

a.     text on lines 56 and 65 is very similarly formulated and is almost perceived as repetition.

Response: Thank you – we combined the two sentences and put the sentence at the beginning of the section.

2: MATERIALS AND METHODS

The method part follows a structured form and where figure 1 in a good way helps the reader to follow the different steps for the overarching project. Hence it is a bit difficult to clearly understand what the current manuscript is focusing. Further, to be able to understand how the development and test of the SENS-Structure has been done according to methodological as well as scientifically-ethical aspects. In the latter, how participants were independent from the researchers and had a voluntary participation. Below are presented some parts that may need to be clarified to understand this as a reader.

Response: We adapted Figure 1 to highlight the phases that are the focus of this article. We are not reporting in this article on the results of phases 3 and 4 so I will not include any other details on studies that were or are going to be performed in these phases.

a.     Pre-clinical phase: here´s a literature search presented as the basis for identifying clinical approaches of thematic structures for care-planning conversations. How this literature review was carried out is not clearly described and in the results section, e.g. the WHO's definition of palliative care and NCCN's guidelines is presented. The question arises whether the authors identified these based on the literature review or basically used well-known documents and guidelines. The discussion then argues that most existing guidelines and documents are not designed for clinics. In fact, I can agree, but wonder if it really is its goal. Here, structured aids in the form of the SENS-structure can be good aids. My wonder is how did the authors reach the guidelines and documents you use? Why, for example, is not The European School of Oncology´s definition of palliative care, or the EAPCs, not included? If those presented in Table 1 were identified in a certain way this needs to be described.

Response: The pre-clinical phase and phase 1 were part of a hospital wide quality-improvement project without the need for ethical approval.
We added the information that the literature search was performed in 2007/2008 under the search terms “palliative care”, “needs assessment”, “assessment of health care needs”, “guideline health planning”, “advance care planning” using the Boolean operators “and/or” respectively.
The ESMO and EAPC definition of PC are both mostly based on the WHO definition and do not serve as assessment structure for clinical use as such.

b.     In this pre-clinical phase, a local interprofessional PC expert team evaluated the thematically structures for its usefulness and practicability. To be able to evaluate the process for me as a reader this teams expertise needs to be described. Is this the same team described in line 58? And presented as “we” in Phase 1? And further professionals in line 99? It needs to be clarified who the expert team are and further who the professionals are.

Response: We added additional information to describe the expertise of the team “The aforementioned professionals, working in a hospital-based specialized PC-consultation service in St.Gallen and Bern, consisted of two physicians and two nurses all trained in PC on specialist level. In addition, a sociologist worked during the time of the development of the structure on her PhD in Palliative Care including communication and interaction with people at the end of life.” This information was added and throughout the text I refer to “our team”.
We added information on who was meant by “professionals” who provided us with feedback in phase 2.

c.     Phase 2: the process within this phase needs to be clarified. As it is described now it poses questions of how this part was performed. Who were involved, including both collaborating PC consultant services? Which were the different settings? How many times were the SENS-structure used? And how was the feedback given? These aspects remain unclear also reading the results from this phase.

Response: Because the ‘SENS’-structure was published in 2012 and since then taught in PC educational trainings as well as in basic medical training, ‘SENS’ was freely available in the Swiss Health care settings and professionals therefore were able to use ‘SENS’ freely in any clinical setting. It is therefore unknown how often SENS was used in the Swiss context but it was tested in the clinical setting (see reference 30) and recommended by the Swiss Federal Office for Public Health (see reference 19 and 31) and accepted as basic assessment structure for Palliative Care by health care insurers (see reference 34).

d.     Phase 3 and 4 are presented briefly and that they should be presented in other publications. Despite this, parts from these phases are included in the results section and with an online survey that is not presented in the method section, which makes it difficult to really understand what the current manuscript focuses on.

Response: We shortened this part to emphasize more on the other phases but not forget all the phases of the MRC framework. The focus of this article is on the pre-clinical phase as well as phases 1 and 2. Results from the studies of phases 3 and 4 were taken out.

3: RESULTS

The result is presented in continuous text and tables in a structured way. However, the same things that are asked for above remains unclear also here.

1.     The entire manuscript wins on the fact that the aspects already mentioned are considered and clarified and then the result might become clearer. It is of importance that it is made clear, who has been involved and which health care facilities, in order to be able to weigh the discussion against results and the development of the SENS-structure. Phase 3 focused on pts with advanced cancer and this is to my view the only time pts diagnosis is described except from descriptions in the introduction. I know that the specific diagnosis might not be the focus for Palliative care, but as the authors focusing om pts with advanced cancer in phase 3, not mention others or how this is handled, and the conclusions made in line 222-223 I think this need some explanation.

Response: We added references to strengthen this conclusion. So far only oncology societies promote the term “concurrent care” and hopefully in the future this perspective will be implemented into the care for patients with other chronic life-limiting diseases. We don’t report on the results of the studies we are conducting in phases 3 and 4.

2.     In Table 1 the authors use abbreviations (Sy, E, N, and SU) and numbers in relations to these abbreviations in some places but not in others. It is not clear how to understand these according to the explanation below the table and this needs to be clarified.

Response: Thank you for your comment on the table – we changed the column of the ‘SENS’ to make it clearer which themes are covered within the different parts of the ‘SENS’- structure and we deleted the abbreviations and wrote the words into the table.

3.     In the part starting at line 147, the authors describe a review made of documentation of clinical conversations. I can’t find any explanation how this was performed. Sometimes the documentation might miss different aspects so it would be helpful to have some descriptions about how this was done.

Response: Dear reviewer, thank you for your important input. Because we systematically reviewed more than 500 documented inpatient consultation documents and analyzed the content within hospital quality improvement project, we can assume that we covered most of the issues patients - disregarding their medical diagnosis - in palliative care could have. To confirm that we are conducting the randomized controlled trial with the embedded qualitative study which will be published later.

4.     Please check line 161 to165 if correctly written.

Response: Thank you – we corrected this paragraph.

5.     The results from phase 4, starting at line 172 describes the SENS- structure to be helpful for inpatient PC and consultant services, and simple to use in various health care settings. It is a bit difficult to value this as no other patient groups but pt with advanced cancer are described and which different health care settings? This needs to be clearified.

Response: This is also a result of the 2nd phase. Within this phase the disease of the patient was not recorded because it was not relevant for the development of ‘SENS’. We changed the listing of the results of phase 2 into A-D to not confuse them with the phases.

4: DISCUSSION

The discussion points out development and use of the SENS-structure covering several years of work, including how the SENS-structure has been developed, implemented in further research and PC curricula at both undergraduate and postgraduate levels, as well as officially acknowledge for use in hospital settings in Switzerland as the basic assessment-structure for ePC. In the earlier parts I have tried to point out some uncertainties that need to be clarified so that readers can form an idea of the secured process in the development and use of the SENS structure. It may well be that the authors who have the results of the two RCT studies clearly understand this, but I do not find it quite clear in this manuscript. Perhaps a larger part of the manuscript should be concentrated on the pre-clinic phase and to include a clear description about in which facilities and for whom the SENS-structure has been tested. Further, I cannot find anything in the result part about test of the SENS-structure in any educational situation although this has already been implemented in PC curricula as well as acknowledge in the Swiss healthcare system as a basic assessment-structure triggering the comprehensive DRG-codes for complex palliative care treatment.

Response: We added quite a few topics in the result section to support the discussion on the potential impact on education and finances. We hope that the additional information throughout the article will help to clarify what we did and how we came to the ‘SENS’- structure.

I hope my comments is clear and to some help to advance the manuscript further.

Response: Yes, your comments were very helpful – thank you again for your clear feedback!

New references: 4, 5, 6, 18, 19, 21, 24, 29-34, 38, 42, 43, 45 and 46
Deleted reference: 29

Reviewer 3 Report

The SENS, as a simple and complete tool, was constructed to proactively plan the best personalised holistic end of care treatment, with the participation of the interprofessional health team, the patients, and their family. Additionally, I congratulate the authors to establish distress as the primary outcome as it is highly prevalent in palliative patients and is negatively related to their quality of life.

However, it seems to me your paper could benefit in clearly emphasise the steps you followed in the pre-clinical phase, phase I and phase II. Phase 3 results will be published in another paper and phase 4 is still underway.

In line 208 you say you «developed SENS structure based on literature, clinical expertise and patients’ needs»; this clear description could come sooner in the text and then explain those steps.

Line 89:

«Common themes served as a first structure. »

To situate the reader, could you describe this first common themes?

Line 94:

The reference [15] was written in German; therefore, it cannot be appreciated by the common reader; I think the article would gain if you could further clarify phase I steps (in point 3.2).

3.1. Pre-clinical phase:

In the literature review, you might describe the strengths and limitations of the three major thematic structures for ePC conversations (WHO, NCCN, GSF). Although you put pertinent information on table 1 some readers may not be acquainted with those tools and a sharper description could benefit the paper.

Additionally, you could refer sooner in the text (you do it on lines 200 and 206), the lack of a simple but at the same time complete tool that could be used by all (interprofessional health team, patient and family) in ePC.

3.2. Phase 1: Piloting and modelling phase

Further clarify the methodology – give examples of the consultations contents related to the main clusters of themes.

3.3. Phase 2: Exploratory phase:

To evaluate the usefulness of the SENS structure, how did you assess feedback from professionals (oral/written form; questionnaire/comments) or did you review documentation of clinical conversations or both? How did you measure? How many professionals/ patients were enrolled in this evaluation? Did you also appraise family members feedback? Please take these questions as an attempt to arouse clarity and gain in your work. 

Lines 192-196:

Could you please describe with more detail the results of your survey?

Lines 211-213:

«Clinical experience showed that ‘SENS’ supports patients and carers to be empowered for future challenges. In addition, it enables teams, together with patients and families, to prioritize and focus on what is important and manageable in individual situations. »

I suggest, if you agree, to add the word “seems”, as you did in line 217.

SENS is still under evaluation as phase 4 is still underway. Although you undoubtedly felt SENS to be clinically useful to patients and families, the results are not yet disposable (or are available in another paper). Therefore, caution should be made in the assumptions you make in the current paper.

Line 347-348: http://www.nfp67.ch/en/projects/module-1-dying-processes-provision-care/project-347-eychmülle

The web page is not online anymore.

Furthermore, and seeking its best results, it seems to me that the SENS could be a valuable educational tool. Healthcare professionals need to gain communication skills to properly adequate the timing and the manner to address a problematic or intimate aspect of palliative care such as advance directives, emotional and spiritual issues.

Please consider my comments such as a possible aid towards more conciseness in the presentation of your work. Seemingly, SENS can be a tool of major value to many patients and professionals. Thank you.

Author Response

Dear reviewer,

We highly appreciate your comments to help us improve the article. We wrote detailed feedback in the separate pdf file which will hopefully contribute to a better understanding of the article and the development of the 'SENS'-structure for Palliative Care interventions.

Best wishes on behalf of the authors,

Monica Fliedner

The SENS, as a simple and complete tool, was constructed to proactively plan the best personalised holistic end of care treatment, with the participation of the interprofessional health team, the patients, and their family. Additionally, I congratulate the authors to establish distress as the primary outcome as it is highly prevalent in palliative patients and is negatively related to their quality of life.

Response: Thank you for your comments and compliments – we appreciate that!

However, it seems to me your paper could benefit in clearly emphasise the steps you followed in the pre-clinical phase, phase I and phase II. Phase 3 results will be published in another paper and phase 4 is still underway.

In line 208 you say you «developed SENS structure based on literature, clinical expertise and patients’ needs»; this clear description could come sooner in the text and then explain those steps.

Response: We added in the methods and result section more details on the steps that we took to develop ‘SENS’.

Line 89:

«Common themes served as a first structure. »

To situate the reader, could you describe this first common themes?

Response: We added the goals of care in common which were identified and clustered alongside the four goals of the Swiss national PC strategy: a) improving self-efficacy and self-help capacity, b) promoting self-determination by supporting a certain sense of coherence in decision-making, c) ensuring safety in sometimes life-threatening situation and d) assuring support of the encumbered carer including bereavement phase which served as the first structure.

Line 94:

The reference [15] was written in German; therefore, it cannot be appreciated by the common reader; I think the article would gain if you could further clarify phase I steps (in point 3.2).

Response: We added more details of the development within the pre-clinical phase and phases 1 and 2.

3.1. Pre-clinical phase:

In the literature review, you might describe the strengths and limitations of the three major thematic structures for ePC conversations (WHO, NCCN, GSF). Although you put pertinent information on table 1 some readers may not be acquainted with those tools and a sharper description could benefit the paper.

Response: We added more details on these three structures. Unfortunately, we realized that the WHO definition of Palliative Care is not for direct clinical, the NCCN structure highly complex and therefore difficult to apply and the focus of the GSF was at the time of review the outpatient population and community care and considered as highly complex. All structures looked at the challenges from the perspective of the health care professional and not from the patient.

Additionally, you could refer sooner in the text (you do it on lines 200 and 206), the lack of a simple but at the same time complete tool that could be used by all (interprofessional health team, patient and family) in ePC.

Response: We added it earlier in the ‘aim’ section.

3.2. Phase 1: Piloting and modelling phase

Further clarify the methodology – give examples of the consultations contents related to the main clusters of themes.

Response: We added quite a bit of details in this section 3.3. Phase.

3.3. Phase 2: Exploratory phase:

To evaluate the usefulness of the SENS structure, how did you assess feedback from professionals (oral/written form; questionnaire/comments) or did you review documentation of clinical conversations or both? How did you measure? How many professionals/ patients were enrolled in this evaluation? Did you also appraise family members feedback? Please take these questions as an attempt to arouse clarity and gain in your work. 

Response: Dear reviewer, we received feedback from the professionals by the survey that we performed. Documentations were in that phase not systematically reviewed We did not take into consideration any formal feedback from family members yet – that is a good idea for future research.

Lines 192-196:

Could you please describe with more detail the results of your survey?

Response: We added more results of the survey.

Lines 211-213:

«Clinical experience showed that ‘SENS’ supports patients and carers to be empowered for future challenges. In addition, it enables teams, together with patients and families, to prioritize and focus on what is important and manageable in individual situations. »

I suggest, if you agree, to add the word “seems”, as you did in line 217.

SENS is still under evaluation as phase 4 is still underway. Although you undoubtedly felt SENS to be clinically useful to patients and families, the results are not yet disposable (or are available in another paper). Therefore, caution should be made in the assumptions you make in the current paper.

Response: We agree and added or changed the words “seems” (conjunctive) since this is the matter of research in the studies in phase 3 and 4.

Line 347-348: http://www.nfp67.ch/en/projects/module-1-dying-processes-provision-care/project-347-eychmülle

The web page is not online anymore.

Response: Thank you for your hint – I corrected the link in the reference to the active website. Unfortunately the direct link to the English website does not work but you can change it to English once you are on the German website. We apologize for that inconvenience.

Furthermore, and seeking its best results, it seems to me that the SENS could be a valuable educational tool. Healthcare professionals need to gain communication skills to properly adequate the timing and the manner to address a problematic or intimate aspect of palliative care such as advance directives, emotional and spiritual issues.

Response: I added a section on educational and financial benefits of SENS in the result section (D) and therefore left the part in the discussion section.

Please consider my comments such as a possible aid towards more conciseness in the presentation of your work. Seemingly, SENS can be a tool of major value to many patients and professionals. Thank you.

Response: Thank you very much for your valuable comments – we hope e made the corrections to enhance the understanding of the paper.

New references: 4, 5, 6, 18, 19, 21, 24, 29-34, 38, 42, 43, 45 and 46
Deleted reference: 29

Round 2

Reviewer 1 Report

The authors have responded constructively and fully to reviewers' comments, and I thought this was much improved.  

Reviewer 2 Report

Dear authors,

I find that you have considered my comments in a good way and handled several of them. You have developed the manuscript and the content is now much clearer.

Best regards